# Cerebral Microvascular Perfusion Assessed in Elderly Adults by Spin-Echo Dynamic Susceptibility Contrast MRI at 7 Tesla

Elles P. Elschot [1,2] , Walter H. Backes [1,2,3], Marieke van den Kerkhof [1,2], Alida A. Postma [1,2], Abraham A. Kroon [3,4] and Jacobus F. A. Jansen [1,2,5,*]

1 Department of Radiology and Nuclear Medicine, Maastricht University Medical Center+, P. Debyelaan 25, P.O. Box 5800, 6202 AZ Maastricht, The Netherlands; e.elschot@maastrichtuniversity.nl (E.P.E.)

2 MHeNs School for Mental Health and Neuroscience, Maastricht University, Minderbroedersberg 4-6, P.O. Box 616, 6200 MD Maastricht, The Netherlands

3 CARIM School for Cardiovascular Diseases, Maastricht University, Minderbroedersberg 4-6, P.O. Box 616, 6200 MD Maastricht, The Netherlands

4 Department of Internal Medicine, Maastricht University Medical Center+, P. Debyelaan 25, P.O. Box 5800, 6202 AZ Maastricht, The Netherlands

5 Department of Electrical Engineering, Eindhoven University of Technology, De Rondom 70, P.O. Box 513, 5612 AP Eindhoven, The Netherlands

* Correspondence: jacobus.jansen@mumc.nl

**Abstract:** Perfusion measures of the total vasculature are commonly derived with gradient-echo (GE) dynamic susceptibility contrast (DSC) MR images, which are acquired during the early passes of a contrast agent. Alternatively, spin-echo (SE) DSC can be used to achieve specific sensitivity to the capillary signal. For an improved contrast-to-noise ratio, ultra-high-field MRI makes this technique more appealing to study cerebral microvascular physiology. Therefore, this study assessed the applicability of SE-DSC MRI at 7 T. Forty-one elderly adults underwent 7 T MRI using a multi-slice SE-EPI DSC sequence. The cerebral blood volume (CBV) and cerebral blood flow (CBF) were determined in the cortical grey matter (CGM) and white matter (WM) and compared to values from the literature. The relation of CBV and CBF with age and sex was investigated. Higher CBV and CBF values were found in CGM compared to WM, whereby the CGM-to-WM ratios depended on the amount of largest vessels excluded from the analysis. CBF was negatively associated with age in the CGM, while no significant association was found with CBV. Both CBV and CBF were higher in women compared to men in both CGM and WM. The current study verifies the possibility of quantifying cerebral microvascular perfusion with SE-DSC MRI at 7 T.

**Keywords:** spin echo; dynamic susceptibility contrast; microvascular perfusion; ultra-high field; cerebral blood flow; 7 Tesla



## 1. Introduction

The assessment of the blood flow through the cerebrovascular system, also known as cerebral perfusion, is an important clinical measurement. Cerebral perfusion measures are clinically mostly used in brain tumors [1], for which a standard protocol has been recommended [2]. For example, malignant gliomas can be differentiated from low-grade gliomas based on the cerebral blood volume (CBV), where higher CBV values are found in high-grade tumor lesions compared to low-grade glioma lesions [3]. Also, the treatment effects and prediction of tumor recurrence after treatment can be assessed using perfusion MRI [4,5]. Another often used clinical application of cerebral perfusion measurements is ischemic stroke [6]. The core and penumbra region of the ischemic lesions can be obtained based on the perfusion–diffusion mismatch concept that is used to select patients who are eligible for endovascular reperfusion therapy [7–9]. Also, the effects of recanalization and reperfusion after endovascular therapy can be assessed with perfusion MRI, gaining

information about the patients' clinical outcome [10]. In addition to the diagnostic value, perfusion MRI is often used for clinical research into various pathologies, providing information on underlying alterations in the microvasculature and subsequent changes in function, for example, in neurodegenerative disorders, where a reduction in perfusion is observed related to cognitive decline [11]. Perfusion MRI can, therefore, be used to bridge the gap between vascular alterations and functional outcomes.

In current clinical use, cerebral perfusion is measured with gradient-echo (GE) dynamic susceptibility contrast (DSC) MR images acquired at 1.5 or 3 Tesla (T). By tracking the CA-induced susceptibility changes during the passage of a contrast agent (CA) bolus caused by the in- and out-flowing blood, regional quantitative perfusion measures of the total vasculature can be obtained, such as the CBV and the cerebral blood flow (CBF) [12]. The conventional GE-DSC sequence is sensitive to signals derived from both the large and small blood vessels, providing a complete view of the cerebrovascular system. Alternatively, the use of spin-echo (SE) DSC MRI has been shown to be specifically sensitive to signal changes in the microvasculature (up to ~10 μm diameter) [13–15], mainly due to the susceptibility-weighted suppression of the macrovasculature. This makes it possible to obtain the specific capillary blood volume and flow, which originates from only 34% of the total vascular bed [16], enabling one to further dive into the physiology at a more microvascular level.

As the onset of neurovascular diseases is often located in the small vessels of the brain, specifically assessing the cerebral microvasculature is of high value to study pathophysiology in early disease assessment or progression monitoring, for example, in cerebral small vessel disease, which has been proven to be a risk factor for dementia and stroke [17]. The use of SE-DSC to specifically assess the microvasculature could, therefore, have a variety of applications in various neurological and neurovascular diseases.

In general, the CA-induced susceptibility changes are smaller in SE compared to GE sequences, due to its specific sensitivity to dynamic field inhomogeneities instead of the combination of dynamic and static field inhomogeneities [14]. To achieve an equal contrast-to-noise ratio (CNR) with an SE sequence, higher CA doses would be required as compared to a GE sequence. With the development of ultra-high-field MRI ($\geq$7 T), the use of SE-DSC MRI becomes more attractive, as the $T_2$ of blood shortens with increasing field strengths. This gives rise to a decrease in the intravascular signal contribution, whereas a higher signal intensity can be measured from the capillaries situated in the tissue. This results in an increased CNR at ultra-high-field strengths without the need to increase the CA dose [18]. Additionally, in SE images, the extravascular (blooming) effects and susceptibility artifacts near air–tissue interfaces are less apparent than in GE images [19,20], improving the effective image resolution. Therefore, SE-DSC at ultra-high-field may provide an interesting alternative to GE-DSC to gain information about the microvasculature, without necessarily having to sacrifice the CNR.

To our knowledge, the use of SE-DSC MRI at ultra-high field to measure cerebral perfusion has not yet been reported. To address this gap, the current study therefore aims to investigate the applicability of SE perfusion at 7 T to assess cerebral microvascular hemodynamics in elderly adults.

## 2. Methods

### 2.1. Study Subjects

Forty-one elderly adults were included in the study (Table 1). As described previously [21], the participants were recruited between November 2019 and June 2021 through local advertisements and via a recruitment website (hersenonderzoek.nl, accessed on 14 January 2024). Exclusion criteria were a history of cerebrovascular disease, transient ischemic attack less than 3 months ago, diagnosed dementia, diabetes mellitus, BMI > 32 kg/m$^2$, and the inability to undergo 7 T MRI. This study was approved by the local Medical Ethical Committee of Maastricht University Medical Center, followed the ethical guidelines of the Dutch Medical Research Involving Human Subjects Act (WMO), and was in line with

the Helsinki Declaration of Human Rights. This study was registered at trialregister.nl (ID: NL7537, date of registration: 20 February 2019). Subject characteristics were recorded, including age, sex, mean arterial pressure, and the Fazekas score (Table 1). Rating of the Fazekas score was performed by a trained neuroscientist (MvdK > 2 years of experience) under the supervision of an experienced neuroradiologist (AAP > 20 years of experience). White matter hyperintensity severity was scored ranging from 0 to 3 in the periventricular white matter (0 = absent, 1 = pencil-thin lining, 2 = smooth halo, 3 = irregular periventricular signal extending into the deep white matter) and in the deep white matter (0 = absent, 1 = punctate foci, 2 = beginning confluence, 3 = large confluent areas) [22]. Written informed consent was provided by all subjects before study participation.

**Table 1.** Participant demographics.

| Characteristic | N = 41 |
|---|---|
| Age (years), median (Q1–Q3) | 65 (58–72) |
| Male, n (%) | 21 (51) |
| Mean arterial pressure (mmHg), median (Q1–Q3) | 93.5 (84.8–99.3) |
| Fazekas score, 0/1/2/3, n (%) | 25 (61)/13 (31.7)/2 (4.9)/1 (2.4) |

*2.2. Image Acquisition*

MR images were obtained using a 32-channel phased-array head coil at 7 T (Siemens Healthineers, Erlangen, Germany). $T_1$-weighted MP2RAGE (TR/TE = 5000/2.47 ms, TI = 900 & 2750 ms, flip-angle = 5 and 3°, acquisition matrix = 224 × 224 mm, cubic voxel size = 0.7 mm, bandwidth = 250 Hz/pixel, acquisition time = 8:00 min:s) and $T_2$-weighted FLAIR (TR/TE = 8000/303 ms, TI = 2330 ms, flip-angle = 120°, acquisition matrix = 192 × 192 mm, cubic voxel size = 1 mm, bandwidth = 383 Hz/pixel, acquisition time = 6:59 min:s) sequences were acquired for anatomical reference. A contrast pre-load was administered at least 30 min prior to the perfusion acquisition (3 mL 1M Gadobutrol, 20 mL saline flush, 0.3 mL/s). During the administration of a second contrast agent bolus (7 mL 1M Gadobutrol, 20 mL saline flush, 5 mL/s), a multi-slice SE EPI DSC-MRI sequence was applied (TR/TE = 3950/60 ms, flip-angle = 90°, acquisition matrix = 96 × 96 mm, 31 slices, slice thickness = 3.5 mm, cubic voxel size = 2 mm, bandwidth = 2084 Hz/pixel, phase-encoding direction = AP, 60 volumes (6 pre-contrast), acquisition time = 4:17 min:s) covering the entire cerebrum. Subsequently, the same sequence in the opposite phase-encoding direction was obtained (TR/TE = 3950/60 ms, flip-angle = 90°, acquisition matrix = 96 × 96 mm, 31 slices, slice thickness = 3.5 mm, cubic voxel size = 2 mm, bandwidth = 2084 Hz/pixel, phase-encoding direction = PA, 5 volumes, acquisition time = 0:40 min:s), to allow for distortion correction.

*2.3. Data Analysis*

2.3.1. Segmentations

$T_1$-weighted MP2RAGE and $T_2$-weighted FLAIR images were used for automatic segmentation of the cortical gray matter (CGM) and white matter (WM) regions, which was conducted using Freesurfer (v6.0.5) [23] and followed by manual corrections when required. Thereafter, region masks were co-registered to the DSC image space using FSL flirt [24].

2.3.2. SE-EPI DSC-MRI

The obtained SE-DSC images (Figure 1) were corrected for head displacement (FSL mcflirt [24]) and EPI distortions (FSL topup [25]). The change in transverse relaxation rate,

which is proportional to the CA concentration, was calculated from the signal changes in the images induced by the CA bolus according to Equation (1).

$$C(t) \propto \Delta R2 = -\left(\frac{1}{TE}\right) \cdot \ln\left(\frac{S(t)}{S(0)}\right) \tag{1}$$

where $C(t)$ is the concentration of CA at a certain timepoint t, $\Delta R2$ is the change in transverse relaxation rate, TE is the echo time, $S(t)$ is the signal at a certain timepoint $t$ and $S(0)$ is the averaged signal before the contrast arrival, which was assumed to be zero.

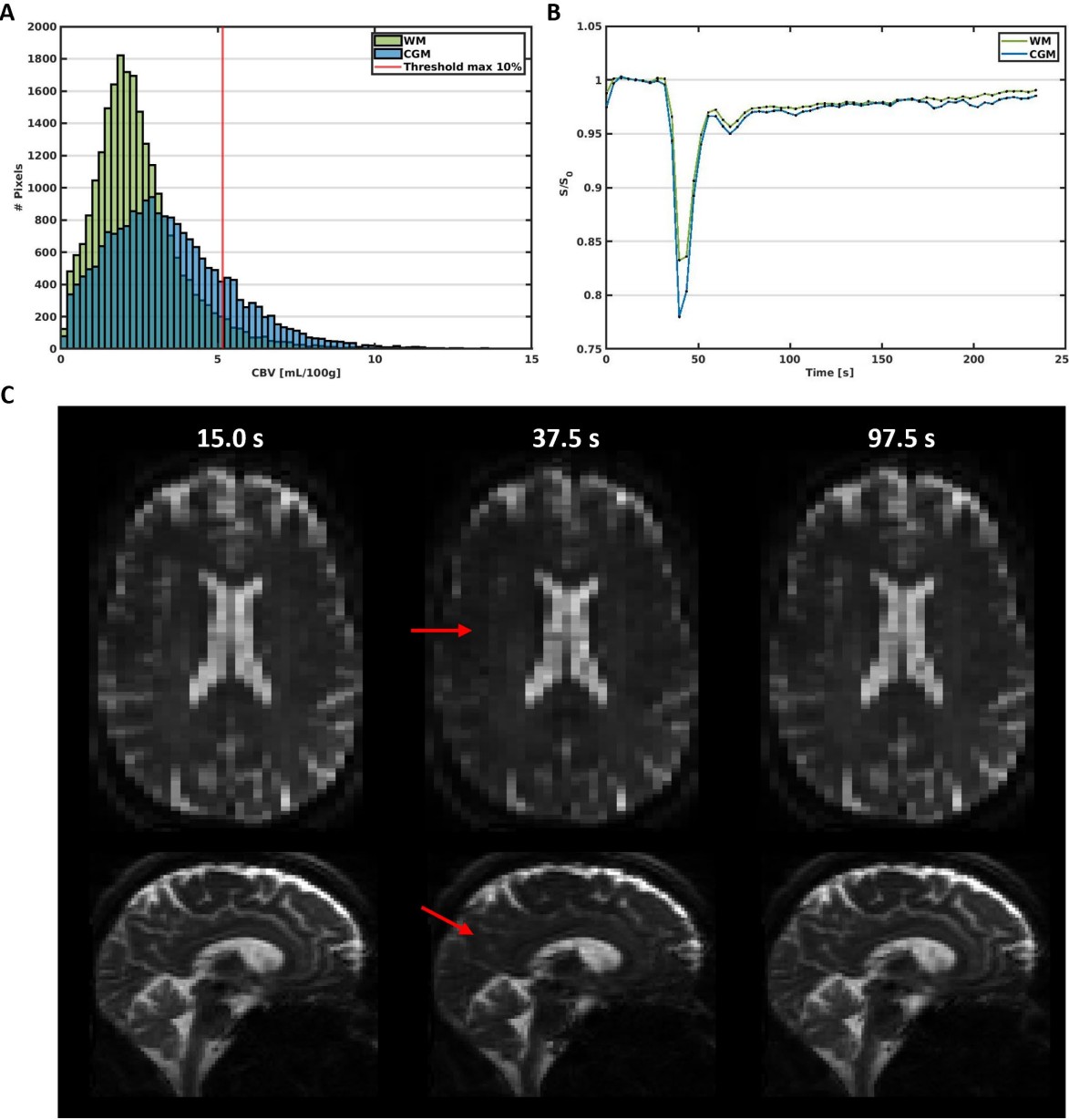

**Figure 1.** (**A**) Histogram of the CBV in the WM and CGM in a representative subject (male, 65 y). The red line represents the 10% threshold, which excludes the large vessels above this threshold in the further analysis. (**B**) Average signal curves in the WM and CGM, showing the signal decrease due to the circulation of the contrast agent bolus over time. (**C**) Images obtained before (15.0 s), during (37.5 s), and after (97.5 s) the administration of the contrast agent bolus. The red arrows point towards areas with visible signal decrease due to contrast agent uptake.

Next, the voxels representing the arterial input function (AIF) were automatically selected in the area of the small vessels branching from the middle cerebral arteries (Figure 2A), based on peak height and time-to-peak signal curve characteristics [26]. After manual corrections, the approximately 5–10 remaining voxels were overlaid on the concentration maps and averaged to obtain the AIF concentration–time curve (Figure 2B). To focus on the first pass of the CA bolus and to correct for the measured recirculation in the analysis, a gamma variate function was fitted through the concentration–time curves [27].

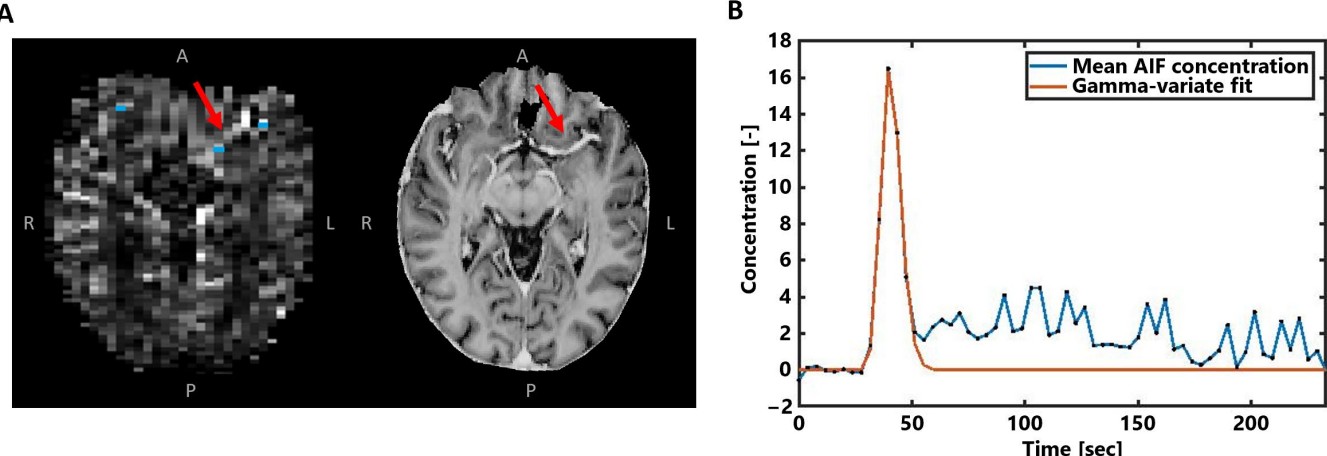

**Figure 2.** (**A**) Example of the AIF voxel selection (blue) visualized with the obtained CA concentration map in the same subject as in Figure 1. The MP2RAGE UNI image on the right is shown for better visualization of the middle cerebral artery highlighted by the red arrows. (**B**) The averaged concentration-time curve of the selected AIF voxels in blue, where the gamma-variate fit is shown in orange.

Thereafter, the CBV was obtained according to Equation (2), using Simpson's rule for integration. Moreover, the CBF was obtained according to Equation (3), using a block-circulant SVD singular value decomposition method [28]. All analyses were performed in a voxel-wise matter.

$$\text{CBV} = \frac{\int_0^\infty C(t)dt}{\int_0^\infty C_{\text{AIF}}(t)dt} \tag{2}$$

$$C_{\text{t}}(t) = \text{CBF} \cdot R(t) \otimes C_{\text{AIF}}(t) \tag{3}$$

where $C_{\text{t}}(t)$ is the concentration of CA in tissue at a certain timepoint $t$, $C_{\text{AIF}}(t)$ is the concentration of CA in the AIF, and $R(t)$ is the tissue residue function.

Subsequently, CBV and CBF maps were scaled to quantitative levels using a global scaling factor based on normalization of the perfusion measures of all subjects in the WM to a reference value obtained with $^{15}$O-PET of 2.7 mL/100 g for the CBV and 22.2 mL/100 g/min for the CBF [29].

Signal contamination of the remaining largest blood vessels was excluded from the analysis by eliminating voxels with the 10% highest CBV values [30] (Figure 3B,C). To investigate the effect of this specific large vessel exclusion threshold on the CBF results, a sensitivity analysis was performed, whereby the removal of the voxels with the highest 0–50% CBV values, roughly representative of the largest vessels, was taken into account. Mean CBV and CBF values were obtained in the different tissue types using the segmented CGM and WM maps. All analyses were performed using custom-made code developed in MATLAB (R2020a, MathWorks, Natick, MA, USA).

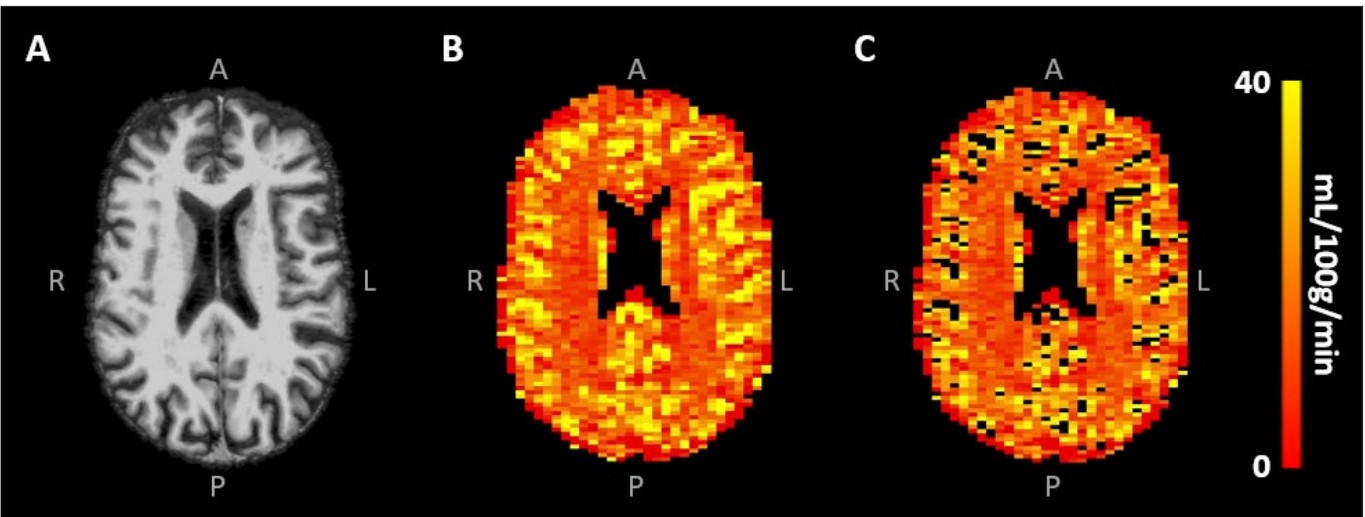

**Figure 3.** (**A**) Skull-stripped T1-weighted image and (**B**) CBF map of the same representative subject as in Figure 1, showing the higher CBF in CGM compared to WM. Removal of the largest vessels (likely veins) occurred mainly in the sulci (**C**).

### 2.3.3. Statistics

Mean CBV and CBF ratios between CGM and WM were compared with ratios from the literature [31,32]. Furthermore, a one-sample t-test was performed to test whether the CBV and CBF ratios between CGM and WM were significantly different from 1. To explore the effect of large vessel exclusion on the CGM-to-WM CBF ratios, a robustness analysis was performed, whereby this analysis was also performed after 0%, 10%, 20%, 30%, 40%, and 50% of the voxels with the highest CBV were excluded from the analysis. Additionally, multivariable linear regression analyses were performed to assess the relation of CBV and CBF with age and sex. All statistical analyses were performed in IBM SPSS (version 27.0, Armonk, NY, USA), where a *p*-value < 0.05 was assumed to be statistically significant.

### 3. Results

Representative SE-DSC images are shown in Figure 1, showing a clear CA bolus peak followed by a smaller recirculation peak and stronger signal drop in the CGM compared to the WM. CBV values were found to be 3.37 $\pm$ 0.90 mL/100 g in CGM and 2.64 $\pm$ 0.68 mL/100 g in WM, showing a higher CBV in CGM compared to WM (CGM-to-WM CBV = 1.28 $\pm$ 0.11, *p* < 0.001; please note that values were scaled to WM literature values [29]). Similarly, CBF values were found to be 31.0 $\pm$ 9.6 mL/100 g/min in CGM and 22.4 $\pm$ 6.6 mL/100 g/min in WM (Figure 3), showing a higher CBF in CGM compared to WM (CGM-to-WM CBF = 1.38 $\pm$ 0.15, *p* < 0.001; please note that values were scaled to WM literature values [29]). The robustness analysis showed a steeper decay of the CGM-to-WM CBF ratio when the 0–10% largest vessels were included in the analysis, compared to removing more large vessels from the analysis. Varying the amount of largest vessels to be removed from the analysis did not change the significance and direction of the associations with age and sex (Figure 4 and Supplementary Table S1).

CBF in the CGM showed a significant negative association with age (Std. $\beta$ = −0.304, *p* = 0.037), while no significant association was found with CBF in the WM (Std. $\beta$ = −0.151, *p* = 0.321). No significant association was found between CBV and age (CGM: Std. $\beta$ = −0.228, *p* = 0.119; WM: Std. $\beta$ = −0.103, *p* = 0.498). Both CBV and CBF were significantly higher in women compared to men in both tissue regions (CBV—CGM: Std. $\beta$ = 0.442, *p* = 0.004; WM: Std. $\beta$ = 0.379, *p* = 0.016; CBF—CGM: Std. $\beta$ = 0.428, *p* = 0.004; WM: Std. $\beta$ = 0.366, *p* = 0.020). The relation between CBF with age and sex is visualized in Figure 5.

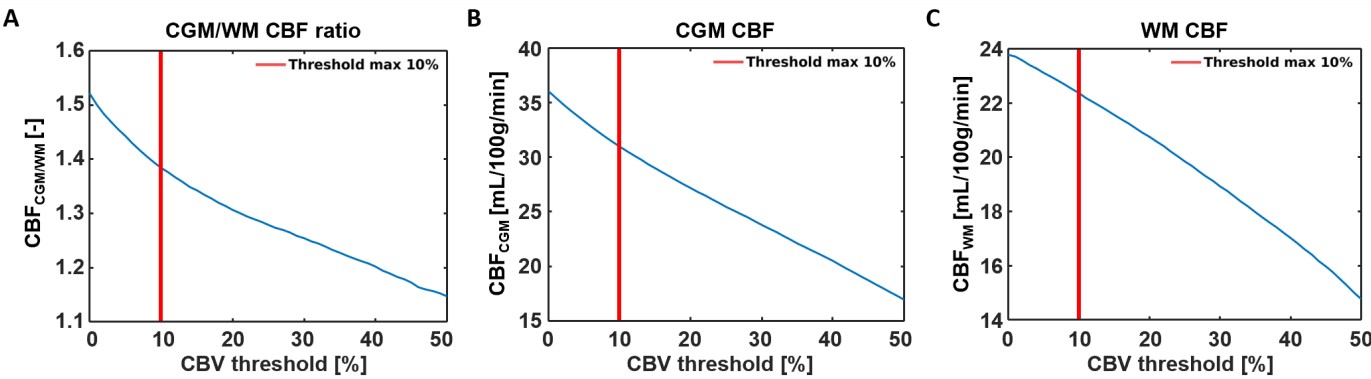

**Figure 4.** Sensitivity analysis showing the effects of removing 0–50% of the large vessels from the CBF analysis. (**A**) A steeper decay of the CGM-to-WM CBF ratio was found when the largest 0–10% were included in the analysis, compared to removing 10% or more large vessels from the analysis. The decrease of CGM-to-WM ratio seems to be more induced by the CBF reduction in the CGM (**B**) than the CBF reduction in the WM (**C**) when removing voxels with the highest CBV values.

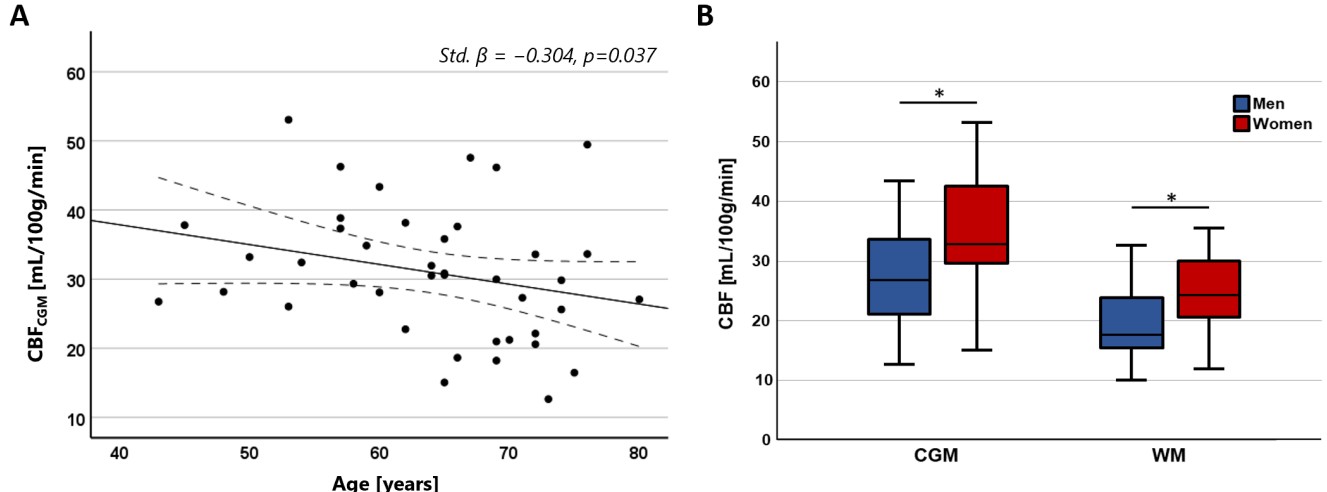

**Figure 5.** (**A**) Scatterplots showing the negative relation between CBF in the CGM and age. (**B**) Boxplots showing higher CBF in women compared to men in both the CGM and WM (* CGM: Std. $\beta$ = 0.428, $p$ = 0.004; * WM: Std. $\beta$ = 0.366, $p$ = 0.020).

## 4. Discussion

The current study demonstrates the first application of SE-DSC MRI at ultra-high-field strength (7 T) to determine the blood perfusion in the cerebral microvessels. Using the SE-DSC, a larger signal decay was observed in the CGM compared to WM, which is indicative of a larger CA uptake (Figure 1). Likewise, the observed CBV and CBF were also found to be higher in CGM compared to WM. Additionally, CBF decreased with age, and both CBV and CBF were found to be higher in women compared to men.

When comparing the values observed in the current study with values from the literature, the higher observed perfusion in the CGM compared to the WM (CGM-to-WM CBF = 1.38 ± 0.15) agreed with previous studies reporting GM-to-WM ratios (GM-to-WM CBV = 1.79–2.6; GM-to-WM CBF = 1.93–3.3) [31,32]. This agreement highlights the validity of our measurement method in a qualitative manner. However, the GM-to-WM ratios found using SE-DSC MRI were 28–58% lower compared to those of previous GE-DSC MRI studies at 1.5T [31,32], based on an intravascular tracer, and 35–54% lower compared to PET and SPECT studies [29,33–37], based on diffusible tracers or microspheres. The discrepancy of the lower perfusion ratios observed in the current study compared to those reported in previous GE-DSC MRI studies might be explained by the specific sensitivity

of SE-DSC sequences to the microvasculature. Considering that most of the larger vessels reside inside or in close proximity to the CGM, the reduced sensitivity of SE-DSC to these vessels might lead to lower CGM values, as measured with GE-DSC, and could, thereby, influence the CGM-to-WM ratio. As a result, a smaller difference between CGM and WM is be observed when SE-DSC is applied, as compared to GE-DSC. The latter statement is further supported by the results of the current study, where the effect of the CBV threshold on the CBF measures was tested (Figure 5) by conducting a robustness analysis. The results of the robustness analysis showed a steeper decrease in CGM-to-WM CBF when including up to approximately 10% of the largest vessels (Figure 5A), thereby supporting the choice to exclude 10% of the largest vessels from this analysis. This effect appeared to be most dependent on the steeper decay of CGM (Figure 5B) compared to WM (Figure 5C), which can be explained by the relatively higher number of large vessels in CGM than WM.

Nevertheless, while PET and SPECT studies do not show the same sensitivity to large vessels as GE-DSC MRI studies, the discrepancy of the higher perfusion ratios compared to the present SE-DSC MRI study still remains. The exact reason why our study at 7 T field strength shows lower GM-to-WM ratios for SE-DSC than those at 1.5T and 3T remains difficult to comprehend. A possible explanation may lie in the differences in the intravascular and extravascular signal contributions for SE-DSC. At 7 T, the T2 value of blood (without gadolinium) is very short (~10 ms), providing no/little signal (see, for example, the AIF of the SE-DSC sequence in Figure 2) in contrast to 3T. This may limit the signal originating from GM (higher microvessel density) more than from the WM (lower microvessel density) due to differences in microvessel density, which could explain why the GM-to-WM ratio is lower at 7 T than at lower field strength. However, other effects may also be involved.

To further validate the SE-DSC sequence as a method to measure perfusion, we investigated the relation of CBV and CBF with variables that were previously known to have a strong established relationship with perfusion within the literature [29,38]. Specifically, previous studies have established associations between CBF and age and sex [29,32,39–42], where CBF declined with older age and was increased in females compared to males. Looking at the different tissue types, the CBF was found to decrease with age in the CGM, which was in line with the literature [29,32,39,41,42]. Regarding the relation of CBF with aging in the WM, contrasting results can be found in the literature. In agreement with some studies [32,42], we found no relation of CBF with age in the WM, while other studies reported a decrease [29,32,41] or even an increase in CBF with age in the WM [39]. Furthermore, CBF was found to be higher in women compared to men, which agreed with previous studies [39–42]. For example, a study by Shin et al. used a GE-DSC MRI at 1.5T [41] and showed an 11% higher CBF in women compared to men in the whole brain. In the current study, an increase of approximately 22% was found in CGM, and an increase of approximately 33% was found in WM in women compared to men. Furthermore, the same study found more pronounced effects of CBF with aging in GM compared to WM (7.4% decrease per decade in GM and 3.0% decrease per decade in WM, considering age-independent average cerebral perfusion values per region). In our research, the CBF decreased approximately 9% per decade in GM and 4.5% per decade in the WM with respect to average CBF values per region. The smaller differences found by Shin et al. compared to our study might be explained by individual scaling to quantitative values, which might filter out some of the age or sex effects. To still be able to measure this variation in this study, a similar global scaling factor was used for all participants. Also, it should be taken into account that a comparison is made with results obtained from GE-DSC. The relation of CBF with age and sex might differ when only taking microvascular CBF into account.

For completeness of this study, we also tested the relation of CBV with age and sex, even though these associations still remain a matter of controversy according to the limited literature that is available [32,41,43]. Contrary to previous studies that did not find an association with sex [32,41], we found an increase in CBV in females compared to males. Furthermore, Leenders et al. found a negative relation of CBV with aging in both GM and

WM tissue types [29], while other studies found a decrease in the GM but no relation in the WM [41,43]. Although the association between CBV and age in this study did show a decline in CGM, this was not significant, in agreement with a study by Bjørnerud and Emblem [32]. A similar study in a larger study population might increase the power and reveal a stronger relation between microvascular CBV and aging.

The main limitation of this study is the lack of a ground truth reference for the obtained CBF measures that are specific to the cerebral microvasculature. Direct comparison with other perfusion techniques (e.g., GE-DSC or PET) would be of value for this study. However, this would also come with its own complications, since it is challenging to obtain truly quantitative measures for SE-DSC. The challenge to derive quantitative measures with SE-DSC might be related to the complicated selection of AIF voxels, due to partial volume effects and the reduced sensitivity to the large inflowing arteries with SE-DSC as a result of short intravascular $T_2$ values. An alternative would be to use more pragmatic (but less physiologically quantitative) perfusion measures that do not use an arterial input function [44]. Furthermore, the direct comparison of SE-DSC with other methods would also come with limitations. For example, the addition of GE-DSC to the already existing SE-DSC protocol would require another, practically not achievable, CA injection. Furthermore, although a comparison of SE-DSC with alternative methods could further validate this technique as a method to measure perfusion, comparisons with other techniques may not be trivial, as different aspects of CBF might be measured. Differences in methodology have been shown to influence GE-DSC perfusion results [2], making it challenging to determine the 'gold standard' GE-DSC method to which the SE-DSC CBF values should be compared. The GE-DSC signal has, for example, been shown to be influenced by blood–brain barrier leakage, which could affect the measured signal [45], and corrections need to be taken into account in the analysis [46]. In this study, we did not include additional corrections in the analysis, as the effect of blood–brain barrier leakage on the SE-DSC signal remains to be investigated. For the current study, we used $^{15}$O-PET literature values as references for the scaling of our measures, as this technique is considered the gold standard for CBF measurements. Nevertheless, it remains unclear to what extent signals from the micro- and macrovessels are captured by this technique, making it also challenging for direct comparison with SE-DSC.

A recently developed method sensitive to vessel size is the combined spin- and gradient-echo (SAGE) DSC sequence [39]. By acquiring SE and GE data simultaneously, the challenges of SE-DSC can be partially resolved. For example, a more reliable AIF can be obtained from the GE data, where a sufficient intravascular signal is present, and a map of the spatial vessel size distribution can be made. Nevertheless, to our knowledge, a realization of this method at ultra-high-field strength (7 T) is currently not yet demonstrated. The use of perfusion phantoms might also be helpful as a gold-standard reference; however, manufacturing perfusion phantoms with microvasculature embedded in soft tissue providing a realistic intra- and extravascular signal is not trivial.

Lastly, the temporal resolution of the SE-DSC sequence used in this study is limited, which might have caused slight under- or overestimation of the perfusion values due to undersampling of the first-pass peak in the concentration–time curves. Simulations performed in earlier research based on a GE-DSC sequence at 1.5T showed underestimated CBF in both GM and WM and reduced GM-to-WM CBF ratios when lowering the temporal resolution. The limited temporal resolution in this study might, therefore, also explain the relatively low GM-to-WM CBF ratios in our study compared to previous studies [47]. The current study was performed using ultra-high-field MRI (7 T), which enabled us to use a lower dose of CA while retaining similar CNR. However, the downside to acquiring data at 7 T MRI is that with increasing field strength, the specific absorption rate (SAR) becomes increasingly challenging [44,48], limiting the temporal resolution in this study. This problem could be overcome by improved sequence design, for example, by using sparse k-space sampling, multiband methods, or acquiring fewer slices, which would further increase the applicability of SE-DSC for perfusion imaging at ultra-high field.

Furthermore, the recent developments in artificial intelligence reconstruction approaches for sparse k-space sampling methods could be of value to increase the temporal resolution of SE-DSC sequences in the future [49], further increasing their applicability.

In addition to optimizing the sequence, there is a need for further research in diseased populations to study the ability of the SE-DSC technique to detect pathophysiological alterations. The particular sensitivity to microvasculature might open a wide range of applications for the use of SE-DSC. For example, it would be interesting to gain more knowledge on the role of microvascular perfusion in disease onset and progression in cognitive decline. Also, the application of SE-DSC might be an advantage in stroke research, where it might help to identify tissue at risk, to more accurately map ischemic damage, to better guide treatment decisions, or to more accurately predict patient outcomes.

In conclusion, the current study demonstrated the application of SE-DSC MRI at 7 T to measure cerebral microvascular perfusion. Our results showed reliable signal–time curves, with a larger signal decrease in CGM compared to WM, corresponding to the literature. Furthermore, SE perfusion decreased with age and was higher in women compared to men, thereby validating this technique with established relations found in the literature. In order to obtain quantitative measures using SE-DSC MRI, future studies should investigate reliable AIF selection, the dependency of R2 relaxivity on CA concentration, and compare the obtained perfusion measures with a ground truth. Future studies in diseased populations should be performed to further evaluate the clinical applications of SE DSC-MRI for pathophysiological purposes.

**Supplementary Materials:** The following supporting information can be downloaded at: https://www.mdpi.com/article/10.3390/tomography10010014/s1, Table S1: Sensitivity analysis investigating the effect of large vessel exclusion on the perfusion results.

**Author Contributions:** Conceptualization, E.P.E., M.v.d.K., W.H.B. and J.F.A.J.; methodology, E.P.E., M.v.d.K., W.H.B. and J.F.A.J.; software, E.P.E.; validation, E.P.E., W.H.B. and J.F.A.J.; formal analysis, E.P.E., M.v.d.K., A.A.P. and A.A.K.; investigation, E.P.E.; resources, E.P.E.; data curation, M.v.d.K. and A.A.P.; writing—original draft preparation, E.P.E.; writing—review and editing, E.P.E., M.v.d.K., W.H.B. and J.F.A.J.; visualization, E.P.E.; supervision, W.H.B. and J.F.A.J.; project administration, J.F.A.J.; funding acquisition, W.H.B. and J.F.A.J. All authors have read and agreed to the published version of the manuscript.

**Funding:** This work is part of the program Translational Research 2, with project number 446002509, funded by ZonMw/Epilepsiefonds.

**Institutional Review Board Statement:** The study was conducted in accordance with the Declaration of Helsinki, and approved by the Institutional Ethics Committee of Maastricht University Medical Center and was registered at trialregister.nl (ID: NL7537, date of registration: 20 February 2019).

**Informed Consent Statement:** Written informed consent was obtained from all subjects involved in the study.

**Data Availability Statement:** Dataset available on request from the authors.

**Acknowledgments:** We are grateful to Chris Wiggins for initial help with the SE-DSC sequence and Esther Steijvers for help with data acquisition.

**Conflicts of Interest:** The authors declare no conflict of interest.

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
