# Peer review of "Cerebral Microvascular Perfusion Assessed in Elderly Adults by Spin-Echo Dynamic Susceptibility Contrast MRI at 7 Tesla"

_tomography, doi:10.3390/tomography10010014_

Round 1

Reviewer 1 Report

Comments and Suggestions for Authors

The authors investigate the suitability of spin-echo dynamic
susceptibility contrast (SE-DSC) MRI at high field strength of 7T for
the study of cerebral blood flow and volume. According to the
authors, no such study is known so far, since investigations are
typically done using gradient echo sequences (I am not in a position
to make a judgment about the novelty of the suggested method).  

The main issue with this manuscript is the comparison to literature
data or any "gold standard". The authors acknowledge that and provide
narrative why this is an intrinsic issue which is complicated to
resolve. I understand why a direct comparison between GE-DSC and
SE-DSC is not feasible (administering a second contrast agent bolus)
but I wonder if a direct comparison could have been done using
perfusion phantoms. While these ma not represent brain matter
correctly they may provide stable sample conditions for direct
sequence comparison.

I suggest a revision of the manuscript also addressing further minor
concerns listed below.    

- The authors use "cerebral perfusion", "cerebral blood volume" and
  "perfusion MRI". The reader may wonder if they mean the same or how
  they are related to each other. More clarity around these terms
  maybe beneficial.

- How is the Fazekas score defined?

- What does MvdK and AAP stand for?

- In section 3 the authors report CGM/WM CBF = 1.4 ± 0.2, while in
  section 4 it is CGM/WM CBF = 1.38 ± 0.15. Why the numbers are
  different?

- This sentence is confusing: "In line with literature[32–35], the CBF
  was found to decrease with age in the CGM and the WM, whilst the
  latter was not significant in our study." I suggest rewording along
  the lines of "In line with literature[32–35], the CBF was found to
  decrease with age in the CGM. However, the same studies also
  reported a decrease of the CBF in WM with age, which was not
  confirmed by our study."

Comments on the Quality of English Language

See "Comments and suggestions for authors".

Reviewer 2 Report

Comments and Suggestions for Authors

1. Main question addressed by the research is  the presentation of a novel SE-DSC method to image perfusion.

2.  The topic is both relevant and original in the field as it is a novel method to acquire perfusion images that has not yet been presented before.

3. What does it add to the subject area compared with other published
material?  

It allows the reader to see a novel method that they might reproduce in future studies.

4.  The methodology is perfectly fine for a technical development paper. A future paper should directly compare this technique with other currently used methods.

5. The conclusion is consistent with the evidence and arguments presented and as far as my expertise is concerned, this method works as described.

6.  References are appropriate and sufficient

7. There is nothing I would change about supplementary, figures, tables, etc.
